# A Comparison of the Anesthetic Methods for Recurrence Rates of Bladder Cancer after Transurethral Resection of Bladder Tumors Using National Health Insurance Claims Data of South Korea

**DOI:** 10.3390/jcm11041143

**Published:** 2022-02-21

**Authors:** Sang Won Lee, Bum Sik Tae, Yoon Ji Choi, Sang Min Yoon, Yoon Sook Lee, Jae Hwan Kim, Hye Won Shin, Jae Young Park, Jae Hyun Bae

**Affiliations:** 1Department of Anesthesiology and Pain Medicine, Korea University Ansan Hospital, Korea University College of Medicine, Ansan 15355, Korea; lsangwon1742@naver.com (S.W.L.); ekha00041@gmail.com (S.M.Y.); yslee4719@gmail.com (Y.S.L.); anejhkim@korea.ac.kr (J.H.K.); 2Department of Urology, Korea University Ansan Hospital, Korea University College of Medicine, Ansan 15355, Korea; jaeyoungpark@korea.ac.kr (J.Y.P.); urobae@korea.ac.kr (J.H.B.); 3Department of Anesthesiology and Pain Medicine, Korea University Anam Hospital, Korea University College of Medicine, Seoul 02841, Korea; drhwshin@naver.com

**Keywords:** bladder cancer, general anesthesia, regional anesthesia, transurethral resection of bladder tumor

## Abstract

Bladder cancers have high recurrence rates and are usually removed via transurethral resection of bladder tumor (TURBT). Recently, some reports showed that the anesthetic method may affect the recurrence rates of bladder cancers. The purpose of this population-based study was to compare the effect of anesthetic methods with the recurrence rates of bladder cancers in South Korea. A total of 4439 patients were reviewed retrospectively using the data of the Korean National Health Insurance (NHI) claims database from January 2007 to December 2011. Patients were divided into 2 groups who received general (*n* = 3767) and regional anesthesia (*n* = 582), and were followed up until September 2017. Propensity score matching was conducted to reduce the effect of confounding factors. After using propensity score matching with a multivariable Cox regression model, age (*p* < 0.001), sex (*p* < 0.001), hypertension (*p* = 0.003), diabetes mellitus (*p* = 0.001), and renal disease (*p* < 0.001) were significantly associated with bladder cancer recurrence. However, there were no significant differences in the recurrence rates of bladder cancers in patients who received general anesthesia and spinal anesthesia for TURBTs. This study revealed that there is no relationship between the anesthetic method and bladder cancer recurrence. Either general anesthesia or regional anesthesia can be used depending on the situation in patients receiving TURBT. Future prospective studies are warranted to confirm the association between the anesthetic method and the recurrence rates of bladder cancer.

## 1. Introduction

Bladder cancer is one of the most frequent cancers diagnosed worldwide and accounts for 3% of cancers worldwide [1]. Approximately 70% of bladder cancers are nonmuscle-invasive bladder cancers (NMIBCs), which can include transurethral resection of bladder tumor (TURBT) for staging and histological diagnosis [2]. NMIBC has a recurrence rate ranging from 15% to 70% per year depending on tumor characteristics, and about 10% progress to muscle-invasive bladder cancer [3]. In general, intravesical BCG (Bacillus Calmette-Guerin) therapy or chemotherapy is recommended after TURBT to reduce the recurrence rate of bladder cancer [4].

As a tumor treatment strategy, it is important to predict tumor recurrence during the follow-up period. According to previous studies [5,6], age, hematuria, number of tumors, tumor size, tumor grade, presence of intraepithelial carcinoma, and intravesical therapy have been investigated as predictors of recurrence of NMIBC. Recently, it has been reported that tumor recurrence after TURBT may be influenced by perioperative factors, such as the type of anesthetic method [7,8]. In general, TURBT is performed under general anesthesia, regional anesthesia, or local anesthesia [9] for patient safety and surgical convenience.

Recently, some studies [7,10,11] have attempted to identify which anesthetic method shows better outcomes regarding the recurrence rate of NMIBC and have reported that patients who underwent TURBT with spinal anesthesia exhibited lower 5-year tumor recurrence rates than that with general anesthesia. However, in another study [10], there was no difference in the 5-year survival rate between general anesthesia (87.5%) and regional anesthesia (96.3%), and the anesthesia method had no significant relationship with the 5-year survival rate.

Therefore, this study attempted to retrospectively investigate the difference between recurrence rates after TURBT for NMIBC patients according to the anesthetic method using data from the National Health Insurance Service of Korea for the last 10 years.

## 2. Method

This study was approved by the Institutional Review Board of Korea University Ansan Hospital (2020AS0131). Information from the Korean national health insurance (NHI) claims database from January 2007 to December 2011 was used. The NHI claims database can provide information on all the insurance claims of the Korean population. The data was supplied after patient de-identification by the NHI.

The study cohort included disease codes diagnosed with bladder cancer (“C67”) among patients who received TURBT (“R3512”). To define newly diagnosed bladder cancer, patients diagnosed with previous bladder cancer or had other cancer history were excluded. Moreover, patients who underwent radical cystectomy, radiotherapy, or chemotherapy due to muscle-invasive bladder cancer were excluded. Lastly, patients who received intravesical treatment (BCG or mitomycin), or who underwent Re-TURBT during the 12 weeks after prior TURBT were also excluded to include only low-risk NMIBC patients.

Baseline characteristics, underlying disease, type of anesthesia, and follow-up data of the study subjects were extracted from the NHI claims database. Patients were divided into the group who underwent general anesthesia (“L0101”, “L1211”) and the group who underwent regional anesthesia (“L1213”).

TURBT (“R3512”), radical cystectomy (“R3481”, “R3482”), intravesical instillation (“R3655”), and intravesical BCG therapy (1143) were the procedure codes used in this study. Mitomycin (1964) was used as the intravesical chemotherapy regimen.

The recurrence of bladder cancer was defined as the case of re-TURBT performed after 3 months of initial TURBT after bladder cancer recurrence was suspected according to cystoscopy or computed tomography (CT). Patients diagnosed with bladder cancer for the last 5 years, from January 2007 to December 2011, were followed up until September 2017. Time to recurrence was defined as the time from the operation date to recurrence.

### Statistical Analysis

Data are represented as the mean ± standard deviation and number of patients (%). The confounding variables and essential characteristics of the two anesthesia groups were analyzed using the independent *t*-test for continuous variables and the Fisher’s exact test or chi-square test for categorical variables.

Propensity score matching was used to evaluate the effect of anesthetic methods on the recurrence rate of bladder cancer and reduce the possibility of potential bias, thereby making the two groups more comparable. After 1:1 propensity score matching was conducted, an independent *t*-test was used to compare continuous variables, and the Fisher’s exact test or chi-square test to compare categorical variables.

The multivariable Cox regression model was used to investigate the relationship of covariates with the recurrence of bladder cancer. The Kaplan–Meier method was used to calculate the cumulative probability of recurrence-free bladder cancer in the total and the propensity score-matching cohort. All statistical analysis was done using SAS^®^ ver. 9.4 (Statistical Analysis Software 9.4, SAS Institute Inc., Cary, NC, USA). If the *p*-value was <0.05, then the data were considered statistically significant.

## 3. Results

The medical records of 20,222 patients who underwent TURBT were reviewed (Figure 1). Due to previous bladder cancer or other cancer history, 9315 patients were excluded. Moreover, 6558 patients were excluded as they had undergone radical cystectomy, radiotherapy, or chemotherapy due to muscle-invasive bladder cancer; received intravesical treatment (BCG or Mitomycin); and underwent Re-TURBT for 12 weeks after prior TURBT. Finally, 4439 patients were included in this study.

Table 1 shows the demographic data of patients who underwent TURBT. Among patients who underwent TURBT, significant differences were observed for age (*p* < 0.001), sex (*p* < 0.001), hypertension (*p* = 0.001), and renal disease (*p* < 0.001) between patients who underwent general anesthesia and regional anesthesia. Finally, 1164 patients (general:spinal = 582:582) were extracted after 1:1 propensity score matching, and the difference between the 2 groups was improved (Table 1). The recurrence rates of both groups showed no significant difference (*p* = 0.617).

Using the multivariable Cox regression model with propensity score matching, age (*p* < 0.001), sex (*p* < 0.001), hypertension (*p* = 0.003), diabetes mellitus (*p* = 0.001), and renal disease (*p* < 0.001) were related to bladder cancer recurrence in patients who underwent TURBT (Table 2). There was no statistical relationship between the anesthetic method and bladder cancer recurrence. Figure 2 shows the Kaplan-Meier curves explaining the time to recurrence of bladder cancer in the general and spinal anesthesia groups. There were no significant differences in the tumor recurrence rates of bladder cancer in the two groups in both the unmatched (Figure 2A) and the propensity score-matching cohort (Figure 2B).

## 4. Discussion

This retrospective study revealed that no correlation exists between the anesthetic method and the recurrence rates of NMIBCs in patients who received TURBT. Choosing which anesthetic method to use is determined by various factors, such as the patient’s preference, perioperative condition of patients, and the decision of the anesthesiologist. In this study, after using propensity score matching and the multivariable Cox model to adjust confounding factors, we found that the type of anesthetic method is not a meaningful variable for the recurrence of bladder cancer.

Researchers have tried to evaluate potential risk factors for predicting the recurrence of NMIBC [5,6,12,13,14]. Several unfavorable prognosis features, such as old age, hydronephrosis, high body mass index, smoking, and tumor characteristics, such as tumor size, multiple tumors, tumor grade, invasion depth, submucosal infiltration, size (>3 cm), high grade, and concomitant tumors in situ, have been associated with a high risk of progression.

In our study, hypertension, diabetes mellitus (DM), and chronic kidney disease were related to recurrence. These factors are known to be possible risk factors for the recurrence of bladder cancer. Teleka et al. [15] revealed that systolic blood pressure was positively related to muscle-invasive bladder cancers (MIBCs). In a propensity score-matching cohort study based on the nationwide population of Taiwan, a positive relationship between hypertension and subsequent urinary bladder cancer possibility was suggested [16].

Furthermore, some studies also identified a relationship between DM and bladder cancer recurrence [17,18,19]. In patients diagnosed with both NMIBC and DM, not taking Metformin may be an important factor for the progression and recurrence of bladder cancer. Inadequate control of glucose seems to be a risk factor for the progression and recurrence of bladder cancer [20]. The mechanism underlying the association between DM and bladder cancer is unclear, but tumor cell proliferation can be induced by chronic exposure to hyperglycemia or hyperinsulinemia [21]. Moreover, cellular proliferation and apoptosis inhibition can be stimulated because the insulin-like growth factor 1 level is increased in DM patients [22].

CKD is also known as a possible risk factor for the progression and recurrence of bladder cancer. Some studies revealed that NMIBC patients who were previously diagnosed with CKD had a poor prognosis compared to other patients [23,24]. The CKD stage was strongly related to worse recurrence-free and progression-free survival [25]. Likewise, the estimated glomerular filtration rate can be used as a significant predictor of the progression and recurrence of NMIBC [26]. The reason why the recurrence rate of NMIBC in CKD patients is higher remains indistinct. One possible suggestion is immune system dysfunction induced by the uremic environment, such as impaired T cell function, decreased B cell count, and macrophage hypoactivity [27]. Moreover, elevated levels of C-reactive protein and the neutrophil-lymphocyte ratio can be found [28]. Another possible mechanism includes reduced DNA repair ability and chromosomal abnormalities [29]. Moreover, oxidative stress can activate chronic inflammation, promoting carcinogenesis and other mechanisms [30].

Recently, some recent studies [7,10,11] have reported that the recurrence rate of NMIBC depends on the anesthetic method. For patient safety and surgery convenience, TURBT generally uses general, regional, and local [9] anesthesia. Jang et al. [10] reported that the 5-year survival rate did not differ between general anesthesia (87.5%) and regional anesthesia (96.3%) and the anesthesia method had no significant relationship with the 5-year survival rate. However, some other reports [7,11] suggested that receiving spinal anesthesia during TURBT for NMIBCs showed a lower recurrence rate than general anesthesia after adjustment for confounding variables and delayed time to recur. Therefore, studies based on big data using appropriate methods are warranted to confirm that the anesthetic method can affect a patient’s surgical outcome. Our results using national claims data revealed that no correlation exists between the anesthetic method and the recurrence rates of NMIBCs in patients who received TURBT.

General anesthesia is a commonly used anesthesia method in TURBT. Because the patient is unconscious and immobile, the operator is comfortable and there is no time constraint. However, drugs, such as volatile drugs and opioids, used during general anesthesia may affect the outcome of TURBT. Volatile agents have been known to have suppressive effects on the immune system, such as T lymphocytes and NK cells. Moreover, it can induce mitogenesis, angiogenesis, and metastasis in cancers [31]. In addition, volatile agents during general anesthesia can stimulate quick production of hypoxia-inducible factor-1 and cancer cell proliferation activation [32]. Therefore, when surgery is performed under general anesthesia in cancer patients, immune escape can happen [33].

Regional anesthesia can minimize the use of opioids and volatile agents that reduce perioperative immunosuppression during surgery. However, patients during regional anesthesia are sometimes conscious and can cough or move, which can affect the surgical procedure. Regional anesthesia is known to attenuate the activation of the hypothalamic–pituitary-adrenal axis and sympathetic nerve activation during the perioperative period [34]. When the hypothalamic-pituitary-adrenal axis is activated, T lymphocytes, NK cells, and macrophage activity can be suppressed. Sympathetic activation during surgery can induce decreased local perfusion and suppress immune cell activity due to effector cells binding with catecholamine [35]. Local anesthetics administered throughout the surgery can more directly affect cancer cells during regional anesthesia because antiproliferative effects of ropivacaine and lidocaine on tumor cells in vitro have been shown [36]. Regional anesthesia can reduce the use of systemic opioids, which can decrease both cellular and humoral immune function [37,38]. It can be related to better oncologic outcomes.

Our study showed different outcomes compared with previous studies [7,10,11]. Our study aimed to analyze data related to the outcomes of bladder cancer patients who received TURBT in a Korean population. The NHI claims database includes all payers, which is the largest management database in Korea. All citizens in Korea are protected by medical insurance from birth. It means that we can use the NHI claims database for conducting research with a large sample size. Using a large sample size, we could confirm the relationship between the anesthetic method and the recurrence of bladder cancer in patients who received TURBT. In addition, our study was conducted with a focus on low-risk NMIBC, which allows for free choice between general anesthesia and regional anesthesia. For high-risk NMIBC, general anesthesia may be required in consideration of surgery. In our results, it was found that general anesthesia was chosen 6.5 times more, which can be attributed to the benefits of hospitals, convenience of anesthesiologists, and consideration of 1-day discharge.

The results of our study showed that the recurrence rate was approximately 18% in both patients under general anesthesia and regional anesthesia. This means that the recurrence rate of NMIBC is not related to the anesthesia method. However, low-risk NMIBC is known as risk of progression is less than 1% in 5 years, risk of recurrence is 32% in 5 years within 5 years, and the longer the period, the higher the progression rate [39,40].

This study followed patients for at least 6 to 10 years and excluded patients with cancer progression; thus, the results of this study may show a low recurrence rate.

Even though our study included data from most of the Korean population, there are some limitations. Our research is a nonrandomized retrospective study, and the data from the NHI claims database are not clinical data and are primarily based on the operation code or disease code. A study that examines the accordance of the diagnosis in NHI data to the actual status of health conditions by comparing medical record reports showed that, on average, 70% of diagnoses correspond to diagnoses in medical charts, although the accordance rate differed depending on the conditions and care setting or types of providers [41]. The number of tumors, tumor size, grade of malignancy, tumor stage, prognostic lifestyle factors, including history of smoking or alcohol intake, etc. are basic characteristics that can affect the research results, but the NHI claims data do not include these data. Our criteria for comorbidities relied on diagnostic codes. Therefore, it is difficult to differentiate the exact stage of cancer. However, according to the NCCN guideline or EAU guideline, for patients with low-risk NMIBC after initial TURBT, surveillance was recommended and intermediate- or high-risk NMIBC patients were recommended re-TURBT within 4–6 weeks after initial TURBT or intravesical treatment (e.g., BCG). Therefore, it can be inferred that patients who received these treatments are intermediate- or high-risk patients. As we only included low-risk NMIBC patients in our study, we excluded patients who underwent re-TURBT or intravesical treatment.

## 5. Conclusions

In conclusion, this retrospective study based on big data extracted from the Korean NHI claims database revealed that the recurrence rates of NMIBCs had no significant relevance with the anesthetic method in patients who underwent TURBT. Therefore, both anesthetic methods can be used depending on the anesthesiologist’s experience and skills and the patient’s condition in NMIBC patients undergoing TURBT; however, more future prospective studies are warranted.

## Figures and Tables

**Figure 1 jcm-11-01143-f001:**
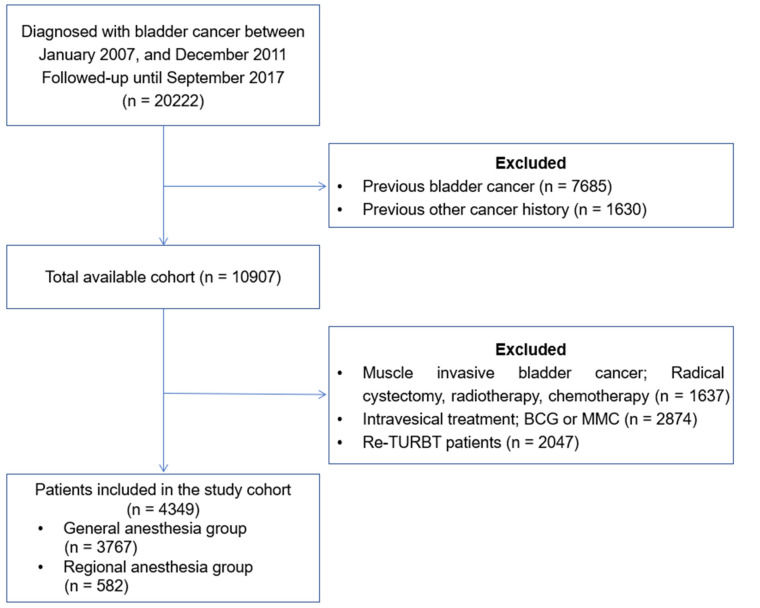
Study flow diagram of the cohort of patients newly diagnosed with low risk of non-muscle-invasive bladder cancer patients in the Korean national health insurance system between 2007 and 2011.

**Figure 2 jcm-11-01143-f002:**
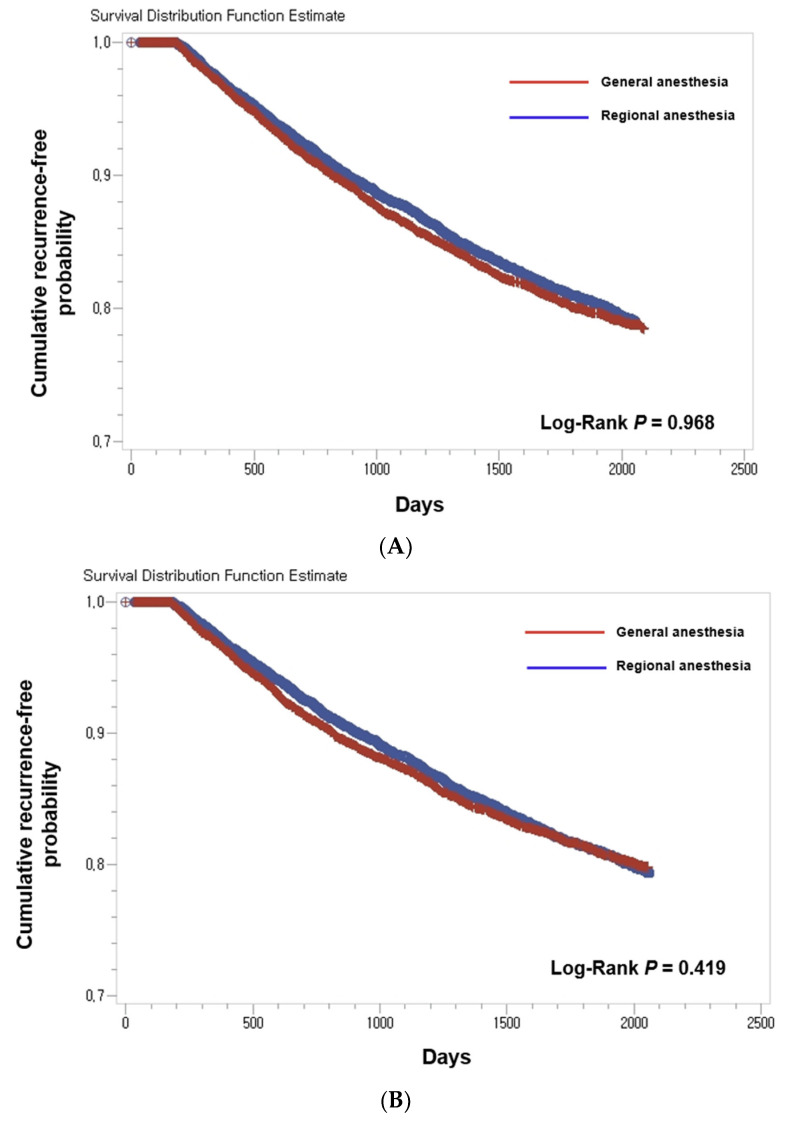
Kaplan–Meier curves of bladder cancer recurrence-free probability. (**A**) in the total cohort (*n* = 4349) and (**B**) in the propensity score-matching cohort (*n* = 1164). Patients who received general anesthesia (red) and who received regional anesthesia (blue).

**Table 1 jcm-11-01143-t001:** Demographic characteristics of patients with bladder cancer, stratified by anesthetic method.

Variable	Full Cohort (*n* = 4349)			Propensity Score-Matched Full Cohort (*n* = 1164)		
	Regional (*n* = 582)	General (*n* = 3767)	*p-*Value	Regional (*n* = 582)	General (*n* = 582)	*p-*Value
Age	66.75 ± 6.09	65.50 ± 6.29	<0.001	66.75 ± 6.09	66.50 ± 6.06	0.094
Gender			<0.001			0.999
Male	460 (79.04%)	3131 (71.99%)		460 (79.04%)	460 (79.04%)	
Female	122 (20.96%)	1218 (28.01%)		122 (20.96%)	122 (20.96%)	
Medical history						
Hypertension	212 (36.45%)	1476 (39.19%)	0.001	212 (36.45%)	212 (36.45%)	0.999
Diabetes mellitus	82 (14.01%)	557 (14.78%)	0.180	82 (14.01%)	82 (14.01%)	0.999
Myocardial infarction	10 (1.65%)	67 (1.78%)	0.519	10 (1.65%)	10 (1.65%)	0.999
Cerebral vascular disease	55 (9.49%)	377 (10.02%)	0.281	55 (9.49%)	55 (9.49%)	0.999
Congestive heart failure	15 (2.52%)	90 (2.40%)	0.660	15 (2.52%)	15 (2.52%)	0.999
Renal disease	30 (5.20%)	311 (8.25%)	<0.001	30 (5.20%)	32 (5.50%)	0.453
Dementia	26 (4.42%)	1772 (4.57%)	0.668	26 (4.42%)	26 (4.42%)	0.999
Liver disease	15 (2.52%)	90 (2.40%)	0.642	15 (2.52%)	15 (2.52%)	0.999
COPD	79 (13.62%)	520 (13.80)	0.766	79 (13.62%)	79 (13.62%)	0.999
Follow-up (day)	1607.05 ± 651.74	1598.93 ± 655.66		1607.05 ± 651.74	1606.19 ± 652.64	
Recurrence	107 (18.38%)	710 (18.85%)	0.451	107 (18.38%)	108 (18.56%)	0.613

Data is presented as the mean ± SD and number of patients (%). SD: standard deviation, COPD: chronic obstructive pulmonary disease.

**Table 2 jcm-11-01143-t002:** Multivariable Cox regression for the association of covariates with bladder cancer recurrence.

Variable	Propensity Score-Matched Full Cohort	
	HR (95% CI)	*p-*Value
Age (≥65)	1.65 (1.47–1.85)	<0.001
Male	1.73 (1.52–1.97)	<0.001
Medical history		
Hypertension	1.24 (1.08–1.42)	0.003
Diabetes mellitus	1.30 (1.13–1.50)	0.001
Myocardial infarction	1.36 (0.89–2.08)	0.161
Cerebral vascular disease	0.65 (0.24–1.75)	0.399
Congestive heart failure	1.04 (0.86–1.27)	0.669
Renal disease	1.74 (1.33–2.28)	<0.001
Dementia	0.95 (0.79–1.13)	0.553
Liver disease	1.01 (0.86–1.19)	0.863
COPD	1.02 (0.91–1.15)	0.725
Anesthetic method		
Regional	1.11 (0.97–1.28)	0.131
General	0.97 (0.79–1.19)	0.743

HR: hazard ratio, CI: confidence interval, COPD: chronic obstructive pulmonary disease.

## Data Availability

Data are not available according to the policy of National Health Insurance Sevice.

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
