# Peer review of "A Comparison of the Anesthetic Methods for Recurrence Rates of Bladder Cancer after Transurethral Resection of Bladder Tumors Using National Health Insurance Claims Data of South Korea"

_jcm, 2022, doi:10.3390/jcm11041143_

Round 1

Reviewer 1 Report

Compared to the previous manuscript, most of the changes indicated by the reviewers, which definitely influenced (positively) the quality of the article. Scientific soundness remains low due to the retrospective nature of the work and limited data that can be obtained from the NHI database.

Some minors remarks:

 - lines 244-245 - "....charlson comorbidity index."  - Should be: Charlson comorbidity index (CII) - wrong spelling. But where there is information about it in the material and methods and results sections? 

 - line 43 - progresses  - maybe  ... 10% of them progress..

 - 

Author Response

 - lines 244-245 - "....charlson comorbidity index."  - Should be: Charlson comorbidity index (CII) - wrong spelling. But where there is information about it in the material and methods and results sections?

  • Thank you for your good comment. We inserted some sentences in discussion.(234-249)

 - line 43 - progresses  - maybe  ... 10% of them progress..

  • Thanks for the comments. We inserted some words.(43)

Reviewer 2 Report

Dear Authors,

your efforts to improve your manuscript are really welcome and fully answered to the points raised from my previous review. Nevertheless, I read thoroughly the report from the other reviewer, especially regarding tumor characteristics which are not fully reported. Low risk NMIBC, according to EAU guidelines, include a primary, single, Ta/T1 LG/G1 tumour < 3 cm in diameter without CIS in a patient < 70 years (exact definition). According to your reported definition of low risk, I considered that you selected just these cases. A further comment in methods about selection and definition of low risk patients used it's mandatory, otherwise readers might be confused and raise doubts on the validity of the manuscript. 

Author Response

your efforts to improve your manuscript are really welcome and fully answered to the points raised from my previous review. Nevertheless, I read thoroughly the report from the other reviewer, especially regarding tumor characteristics which are not fully reported. Low risk NMIBC, according to EAU guidelines, include a primary, single, Ta/T1 LG/G1 tumour < 3 cm in diameter without CIS in a patient < 70 years (exact definition). According to your reported definition of low risk, I considered that you selected just these cases. A further comment in methods about selection and definition of low risk patients used it's mandatory, otherwise readers might be confused and raise doubts on the validity of the manuscript.

  • Thanks for the nice comment. We inserted some sentences in discussion.(234-249)

This manuscript is a resubmission of an earlier submission. The following is a list of the peer review reports and author responses from that submission.

Round 1

Reviewer 1 Report

The topic taken up by the authors is very interesting and intriguing.
Previous studies have indicated the existence of certain dependencies.
Unfortunately, despite the large database, the research thesis is considered very vaguely.
The article is a retrospective work and the database is from the NHI.
The introduction is inconsistent and does not clearly define why and why the authors raise this topic.
Much specific data on the basic characteristics of tumors are lacking:
 - Number of tumors
 - tumor size
 - grade of malignancy LG or HG
 - tumor stage pTa or pT1
 - Smoking history
These are the basic data relating to the risk of tumor recurrence
There is no ASA score analysis that indicates perioperative risk
What was the algorithm for selecting the type of anesthesia?
Propensity score matching was performed 1: 1 - using the potential of the entire cohort why the authors did not use a different 1: 4 ratio for example.
The authors do not explain in the discussion why there is such a large disproportion between regional and general anesthesia. In most similar works the proportions are reversed - this requires some explanation.
The layout of the discussions should also be changed. The discussion of the results of the research thesis should be at the beginning of this part.

Examples of inaccuracies:
 - line 41 - What are advanced neoadjuvant therapies added to TURB ?. Besides this reference [no 3] seems old
 - lines 46-47 - commonly used? A trivial sentence. Only very small, tiny ones are fulgurated under local anesthesia.
 - lines 47-50 - How does this relate to the research topic? The authors investigate changes in the immune response?
 - line 56-57 - Such a sentence should not be in the original work - Who confirms the truth and who questions the value of the work?

To sum up - an interesting topic, but treated too generally, with little specificity in terms of recurrences of bladder cancer. The material presents potential but requires re-analysis taking into account the above remarks.

Author Response

Dear Editor,

Ref: jcm-1509711

Title: A comparison of the anesthetic methods for recurrence rates of bladder cancer after transurethral resection of bladder tumors using national health insurance claims data of South Korea

Journal: Journal of Clinical Medicine

I appreciate to the editor and referees of the “Journal of Clinical Medicine” for taking their time to review our article. We have made some corrections in our manuscript after going over the referee’s comments.

I hope the revised manuscript will better meet the requirement of the “Journal of Clinical Medicine” for publication. I try to have corrected everything you have pointed out. And I will modify the sentences if there are still a few residual problems with my manuscript. Thank you again for the constructive review by referees.

The changes are summarized below:

- Reviewer 1: 

The topic taken up by the authors is very interesting and intriguing.
Previous studies have indicated the existence of certain dependencies.
Unfortunately, despite the large database, the research thesis is considered very vaguely.
The article is a retrospective work and the database is from the NHI.
The introduction is inconsistent and does not clearly define why and why the authors raise this topic.

  • Thank you for your interest in my article. As this is a retrospective study, there are many limitations, but I will do my best to correct the points pointed out.

Much specific data on the basic characteristics of tumors are lacking:
 - Number of tumors
 - tumor size
 - grade of malignancy LG or HG
 - tumor stage pTa or pT1
 - Smoking history
These are the basic data relating to the risk of tumor recurrence

  • Thanks for the good point. Regarding the above items you mentioned, we cannot know the details of the data using our NHI. This was described in the discussion section of the thesis as a limitation.(237-240)

There is no ASA score analysis that indicates perioperative risk

  • Thank you for your deep consideration. A study that examines the accordance of diagnosis in HIRA data to the actual status of health conditions by comparing medical record reports shows that, on average, 70% of diagnoses correspond to diagnoses in medical charts although the accordance rate was different depending on conditions and care setting or types providers. (Kim et al. J Korean Med Sci. 2017 May;32(5):718-728) However, this study had a retrospective design based on claims data, and our criteria for comorbidities relied on diagnostic codes. NHIS database did not include information on prognostic lifestyle factors including history of smoking or alcohol intake or clinical data. Therefore, since the ASA score could not be calculated, we compared the two groups based on the diagnosis included in the charlson comobidity index. However, as your recommendation, we added referenced in Discussion session. (236-239)

What was the algorithm for selecting the type of anesthesia?

  • For high-risk NMIBC, general anesthesia may be required in consideration of surgery. Therefore, this study was conducted with a focus on low-risk NMIBC, which allows for free choice between general anesthesia and regional anesthesia. However, in our results, it was found that general anesthesia was chosen 6.5 times more, which can be attributed to the benefits of hospitals, convenience of anesthesiologists, and consideration of one-day discharge.
  • ….. In this regard, we have added it to the discussion section.(222-224)

Propensity score matching was performed 1: 1 - using the potential of the entire cohort why the authors did not use a different 1: 4 ratio for example.

  • We wanted to remove all the effects of many variables related to recurrence through matching, and since the best model was actually 1:1 matching, we used 1:1 matching. 1:1 ratio between matched subjects is most commonly used. However when the control group includes many more subjects that the intervention group, other ratios may be used. McAfee et al. used a matching ratio of 1:4 for a larger number of control subjects than test subjects in order to improve study power. (McAfee AT, et al. Pharmacoepidemiol Drug Saf. 2006;15:444–453) Since control group (General group) was much larger than regional group, we choose 1:4 propensity matching. Thank you for your consideration.

The authors do not explain in the discussion why there is such a large disproportion between regional and general anesthesia. In most similar works the proportions are reversed - this requires some explanation.

  • We have added it to the discussion section.(222-225)

The layout of the discussions should also be changed. The discussion of the results of the research thesis should be at the beginning of this part.

  • Thanks for the good comments. Overall, the discussion has been corrected.

Examples of inaccuracies:
 - line 41 - What are advanced neoadjuvant therapies added to TURB ?. Besides this reference [no 3] seems old

  • It was considered unnecessary content, so the contents of the introduction were corrected and the reference was replaced.(40-43)

 - lines 46-47 - commonly used? A trivial sentence. Only very small, tiny ones are fulgurated under local anesthesia.

  • It was considered unnecessary content, so the contents of the introduction were corrected and the reference was replaced. We deleted this sentence.

 - lines 47-50 - How does this relate to the research topic? The authors investigate changes in the immune response?

  • It was considered unnecessary content, so the contents of the introduction were corrected and the reference was replaced.(46-52)

 - line 56-57 - Such a sentence should not be in the original work - Who confirms the truth and who questions the value of the work?

  • It was considered unnecessary content, so the contents of the introduction were corrected and the reference was replaced.(46-52)

To sum up - an interesting topic, but treated too generally, with little specificity in terms of recurrences of bladder cancer. The material presents potential but requires re-analysis taking into account the above remarks.

Thank you for your reviewing. We are happy to have a chance of submitting and reviewing our manuscript. And we look forward to publishing our paper in “Journal of Clinical Medicine”.

Reviewer 2 Report

Dear Authors,

your study has a sound methodology and results are clearly presented. Some points might be analyzed further:

1) In introduction, you might cite also one previous work by Brausi et al, proved the feasibility of local anesthesia in bladder cancer treatment DOI: https://doi.org/10.1016/j.eururo.2007.04.086

2) Methods: including only low risk NMIBC might be a selection BIAS. Natural history of low risk NMIBC is to have a good prognosis both on recurrence and survival. Therefore, numbers might be too low to obtain significant results. It would be better to include also intermediate and high risk NMIBC at first diagnosis

3) Methods: anesthesia impact can affect also complications, thus, as a secondary objective, they should be evaluated if feasible

4) Impact of anesthesia on oncological outcomes might also be affected by surgeons'. General anesthesia might allow surgeon to work in relative tranquillity, while regional anesthesia can affect surgical operation (coughing, etc..) thus having an impact on its efficacy. Add this point to discussion

5) Include also risk of selection bias of low risk NMIBC in discussion for the obtained results, as risk of progression is less than 1% in 5 years, risk of recurrence is 32% in 5 years (PMID: 21228821 DOI: 10.1038/nrurol.2010.208), thus with a sample of roundabout 500 patients in each arm, there should be 5 progression in each group and 160 recurrence for each, it might be possible to not obtain significance.

6) The low recurrence rate compared to what expected might be another result to address in discussion (100 vs 160 in each group)

Author Response

Dear Editor,

Ref: jcm-1509711

Title: A comparison of the anesthetic methods for recurrence rates of bladder cancer after transurethral resection of bladder tumors using national health insurance claims data of South Korea

Journal: Journal of Clinical Medicine

I appreciate to the editor and referees of the “Journal of Clinical Medicine” for taking their time to review our article. We have made some corrections in our manuscript after going over the referee’s comments.

I hope the revised manuscript will better meet the requirement of the “Journal of Clinical Medicine” for publication. I try to have corrected everything you have pointed out. And I will modify the sentences if there are still a few residual problems with my manuscript. Thank you again for the constructive review by referees.

The changes are summarized below:

- Reviewer 2: 

Dear Authors,

your study has a sound methodology and results are clearly presented. Some points might be analyzed further:

1) In introduction, you might cite also one previous work by Brausi et al, proved the feasibility of local anesthesia in bladder cancer treatment DOI: https://doi.org/10.1016/j.eururo.2007.04.086

  • Thanks for letting me know about a good thesis, I added it to the introduction section.(52)

2) Methods: including only low risk NMIBC might be a selection BIAS. Natural history of low risk NMIBC is to have a good prognosis both on recurrence and survival. Therefore, numbers might be too low to obtain significant results. It would be better to include also intermediate and high risk NMIBC at first diagnosis

  • Thanks for the good comments. We seriously thought about the design again, and since this study is a large-scale retrospective study, it was concluded that the inclusion of intermediate and high risk NMIBCs in this study may influence the choice of anesthesia method.
  • It was a good proposal, but unfortunately it is difficult to reach the purpose of the study.
  • We add some sentences in discussion section about this.(220-225)

3) Methods: anesthesia impact can affect also complications, thus, as a secondary objective, they should be evaluated if feasible

  • Thanks for the good comments. It would be nice if we could also add data on complications according to each anesthesia method, but we could not collect data because of the limitation that NIH data does not provide data on complications. We are also sorry for this. We add some sentences in discussion section about this.(236-245)

4) Impact of anesthesia on oncological outcomes might also be affected by surgeons'. General anesthesia might allow surgeon to work in relative tranquillity, while regional anesthesia can affect surgical operation (coughing, etc..) thus having an impact on its efficacy. Add this point to discussion

  • Thank you for your kind comments. It has been attached to the discussion section.(201-202)

5) Include also risk of selection bias of low risk NMIBC in discussion for the obtained results, as risk of progression is less than 1% in 5 years, risk of recurrence is 32% in 5 years (PMID: 21228821 DOI: 10.1038/nrurol.2010.208), thus with a sample of roundabout 500 patients in each arm, there should be 5 progression in each group and 160 recurrence for each, it might be possible to not obtain significance.

  • Thank you for your good comments. It has been attached to the discussion section.(226-232)

6) The low recurrence rate compared to what expected might be another result to address in discussion (100 vs 160 in each group)

  • Thank you for your good comments. It has been attached to the discussion section.(226-232)

Thank you for your reviewing. We are happy to have a chance of submitting and reviewing our manuscript. And we look forward to publishing our paper in “Journal of Clinical Medicine”.
